# Identification of Clinical and Genomic Features Associated with SARS-CoV-2 Reinfections

**DOI:** 10.3390/v17060840

**Published:** 2025-06-11

**Authors:** Francisco Muñoz-López, Antoni E. Bordoy, Francesc Català-Moll, Verónica Saludes, David Panisello Yagüe, Mariona Parera, Ignacio Blanco, Pere-Joan Cardona, Cristina Casañ, Ana Blanco-Suárez, Sandra Franco, Álvaro F. García-Jiménez, Roger Paredes, Bonaventura Clotet, Lourdes Mateu, Marc Noguera-Julian, Elisa Martró, José Ramón Santos, Marta Massanella

**Affiliations:** 1IrsiCaixa, Badalona, 08916 Barcelona, Spain; fmunoz@irsicaixa.es (F.M.-L.); fcatala@irsicaixa.es (F.C.-M.); mparera@irsicaixa.es (M.P.); sfranco@irsicaixa.es (S.F.); rparedes@irsicaixa.es (R.P.); bclotet@irsicaixa.es (B.C.); mnoguera@irsicaixa.es (M.N.-J.); 2Northern Metropolitan Clinical Laboratory, Microbiology Department, Hospital Universitari Germans Trias i Pujol, 08916 Badalona, Spain; aescalas@igtp.cat (A.E.B.); vsaludesm.germanstrias@gencat.cat (V.S.); dpanisello@igtp.cat (D.P.Y.); iblanco.germanstrias@gencat.cat (I.B.); pcardonai.germanstrias@gencat.cat (P.-J.C.); ccasan.germanstrias@gencat.cat (C.C.); ablancos.germanstrias@gencat.cat (A.B.-S.); emartro@igtp.cat (E.M.); 3Germans Trias i Pujol Health Sciences Research Institute Foundation (IGTP), 08916 Badalona, Spain; 4Epidemiology and Public Health Networking Biomedical Research Center (CIBERESP), Health Institute Carlos III, 28029 Madrid, Spain; 5Clinical Genetics Department, Northern Metropolitan Clinical Laboratory, Hospital Universitari Germans Trias i Pujol, 08916 Badalona, Spain; 6Genetics and Microbiology Department, Universitat Autònoma de Barcelona, Cerdanyola del Vallès, 08193 Catalonia, Spain; 7Respiratory Diseases Networking Biomedical Research Center (CIBERES), Health Institute Carlos III, 28029 Madrid, Spain; 8Department of Immunology and Oncology, National Centre for Biotechnology, Spanish National Research Council (CNB-CSIC), 28049 Madrid, Spain; af.garcia@cnb.csic.es; 9Infectious Diseases Department, Hospital Universitari Germans Trias i Pujol, 08916 Badalona, Spain; lmateu@lluita.org; 10Fight Against Infections Foundation, 08916 Badalona, Spain; 11Infectious Diseases Networking Biomedical Research Center (CIBERINFEC), 28029 Madrid, Spain; 12Chair in Infectious Diseases and Immunity, Centre for Health and Social Care Research (CESS), Faculty of Medicine, University of Vic-Central University of Catalonia (UVic-UCC), 08500 Vic, Spain

**Keywords:** COVID-19, reinfection, SARS-CoV-2, viral evolution, comorbidities, vaccination

## Abstract

Although SARS-CoV-2 reinfections remain a concern for healthcare systems worldwide, the factors driving them are still not fully understood. In this study, we examined data for 3303 individuals who experienced two SARS-CoV-2 infections between March 2020 and May 2022 from both clinical and viral genomics perspectives. Our findings indicate that viral evolution was the primary driver of reinfection. However, chronic conditions were common among reinfected individuals, including those under 26 years old, suggesting that the presence of underlying and/or chronic conditions increases susceptibility to reinfection. The median time elapsed between infections was one year, often coinciding with the emergence of new variants. While vaccination showed only a limited protective effect against reinfection, it drastically decreased the hospitalization rate, underscoring its role in mitigating disease severity. Our findings point to the need for more flexible vaccination strategies, especially for individuals with chronic conditions. Understanding the interactions between host factors and viral evolution is critical to strengthening prevention strategies and reducing the burden of reinfections and their possible long-term complications.

## 1. Introduction

As the SARS-CoV-2 pandemic evolves, reinfections are becoming increasingly common. Understanding the clinical characteristics and risk factors associated with reinfection, along with viral evolution, may be useful for guiding public health strategies and shaping vaccination efforts.

Early reports of reinfections highlighted the genetic differences between viral strains and distinct viral lineages approximately three months after the initial infection [1,2]. Since then, SARS-CoV-2 reinfections have received increasing scientific attention due to their potential implications for disease severity, vaccine effectiveness, and long-term health outcomes. While COVID-19 vaccines were primarily designed to prevent severe disease rather than infection, they have shown short-term protection against infection, particularly in the months following vaccination [3]. However, this protection decreases over time [4,5], especially with the emergence of new variants. Booster doses have been shown to enhance and extend immunity, reducing the risk of reinfection [6]. Studying reinfections is crucial for understanding their clinical impact, the evolution of SARS-CoV-2, and the potential long-term consequences, including the risk that patients will develop long COVID [7].

This study aims to provide a comprehensive overview of SARS-CoV-2 reinfections, focusing on the clinical characteristics of affected individuals and the role of viral evolution. For that purpose, we retrospectively selected participants with confirmed SARS-CoV-2 reinfection and then documented their clinical timeline, demographics, health status, vaccination history, and reinfection patterns. Additionally, we performed viral sequencing on a portion of viral samples from both primary infection and reinfection to assess the genetic variability of the virus and its potential impact on reinfection dynamics.

## 2. Materials and Methods

### 2.1. Study Design and Participants

We conducted a retrospective cohort study of reinfections using clinical records and SARS-CoV-2 genomic sequencing data for samples (RECOVID study) obtained from the Clinical Laboratory of the Northern Metropolitan Area (Laboratori Clínic de la Metropolitana Nord, LCMN), a referral laboratory located at the Hospital Universitari Germans Trias i Pujol (HUGTiP). This laboratory serves the northern area of Barcelona Province in Catalonia, Spain, and has systematically stored nasopharyngeal or nasal swabs from SARS-CoV-2 tests throughout the pandemic. All samples underwent routine nucleic acid amplification testing (NAAT), including reverse transcription real-time PCR, RT-PCR (Allplex 2019-nCoV assay [RdRp, E, and N genes, Seegene, Seoul, South Korea], Alinity [RdRp and N genes, Abbott, Chicago, IL, USA], Simplexa [ORF1ab and/or S genes, DiaSorin, Saluggia, Italy], and GeneXpert [RdRp, E, and N genes, Cepheid, Sunnyvale, CA, USA]) using CFX96 instruments (Bio-Rad Laboratories, Hercules, CA, USA) or transcription-mediated amplification (TMA) using the Panther System (ORF1ab gene, Hologic or Procleix, Grífols S.A., Sant Cugat del Vallès, Spain) to detect the presence of SARS-CoV-2 infection.

On our request, the HUGTiP’s Information Systems Department (Direcció d’Organització de Sistemes d’Informació, DOSI) identified SARS-CoV-2 reinfections in an anonymized database they manage called CORE-COVID and provided us with the resulting data subset, which we labeled the RECOVID cohort. This cohort consisted of a total of 3303 individuals reinfected with SARS-CoV-2 between March 2020 and May 2022, with at least two positive PCR tests ≥90 days apart. Demographic, clinical, virological, and immunological data from the database were extracted following the European Union’s General Data Protection Regulation (GDPR) to ensure participant anonymity; therefore, no informed consent was required. This study was approved by the HUGTiP’s Institutional Review Board (PI-21-235). All procedures adhered to good clinical practices and complied with GDPR 2016/679 [8] to ensure participant privacy and confidentiality.

Variables recorded included demographic and clinical data (sex, age, comorbidities, adjusted morbidity groups) and SARS-CoV-2-related information (nucleic acid amplification tests [RT-PCR or TMA], rapid antigen tests, hospitalization, symptomatology, serology, vaccination [dates, manufacturer, and number of doses]).

For comparative purposes, health data for the general population of Catalonia (one of Spain’s constituent Autonomous Regions) was gathered from the Modules for the Monitoring of Quality Indicators (MSIQ) platform of the Catalan Health Institute (CatSalut).

### 2.2. Clinical Timeline Encoding

To facilitate analysis, the clinical timeline of each participant was encoded into a structured string format. Each string began with the date of the first recorded event, followed by a sequence of letters and numbers. The letters represented specific events such as SARS-CoV-2 infection, vaccination, and type of vaccine, and NAAT and antigen tests (among other things), while the numbers indicated the number of days between consecutive events. A detailed description of these timeline strings is shown in Appendix A. This encoding was carried out using Python 3.11.

### 2.3. Definitions and Categorization

A summary of the key definitions used in the study is provided in Table 1 to ensure clarity about the criteria used for important variables including reinfection, extended infection, breakthrough infection, partial breakthrough infection, booster doses and Adjusted Morbidity Groups (AMG, a measure of general health status developed by the Catalan Health Institute [9]). In order to study the association between vaccination and reinfection, participants were categorized into four distinct groups based on the timing of their infections (I = infection) and vaccinations (V = first vaccine, in any possible case {Ad26.COV2.S dose, BNT162b2/mRNA-1273/AZD1222 first dose or first + second doses}; B = booster dose), as follows: (1) VII[B], consisting of individuals vaccinated before their first infection without receiving a booster; (2) IVI[B], individuals vaccinated between infections but without a booster; (3) IVBI, individuals vaccinated and boosted between infections; and (4) VIBI, individuals vaccinated before the first infection and subsequently boosted before the second infection. Except for one individual, all participants received a single booster dose.

### 2.4. SARS-CoV-2 Whole-Genome Sequencing

To address the role of viral genomic variability in the reinfection process, we selected a subset of 130 pairs of nasopharyngeal swab samples (first episode from August 2020 to September 2021 and second episode from February 2021 to March 2022). All SARS-CoV-2-positive selected samples initially processed by TMA were subjected to real-time RT-PCR to determine the cycle threshold (Ct) value. Only those samples with a Ct  < 30 for the N gene target were selected for whole-genome sequencing (WGS) to maximize the sequencing success rate. Genomic reverse transcription, tiling PCR amplification, and sequencing were performed as previously described [11]. The ARTIC network v3 amplicon panel (Integrated DNA Technologies, Coralville, IA, USA) was used for amplification, the Illumina DNA Prep kit for library preparation (Illumina, San Diego, CA, USA), and the Illumina MiSeq platform for sequencing.

### 2.5. Bioinformatic and Phylogenetic Analysis

The sequences were analyzed with the nf-core/viralrecon pipeline [12,13], which executes all the steps, from quality control and alignment of raw sequences to SARS-CoV-2 lineage assignment using Nextclade [14] and Pangolin [15]. We discarded 9 sequences due to low quality (<90% coverage), and after some tagging errors, 74 pairs of sequences (one from each episode of the same participant) and 52 unpaired sequences remained for further analysis. All FASTA sequences with good quality were uploaded to the GISAID database [16] (https://www.gisaid.org/; accession ID: EPI_SET_250311pz; accessed on 11 March 2025).

We extracted the spike gene of the paired sequences and used MEGA software version 12 [17] to select a suitable distance model to build a phylogenetic tree. The model with the lowest BIC was T92+G+I. Then, we built the tree in the IQ-TREE [18] web application. The tree was rooted in the reference SARS-CoV-2 sequence (Wuhan-Hu-1/MN908947.3 [19]). The representations of the tree were created using FigTree (http://tree.bio.ed.ac.uk/software/figtree/; version 1.4.4, accessed on 11 March 2025). We focused on the spike gene because it harbors the majority of mutations responsible for immune escape and is the primary target of neutralizing antibodies and vaccines. Analyzing this region provides critical insights into reinfection risk driven by antigenic evolution.

Finally, we devised a genetic distance model to study the importance of several variables in the genetic distance between pairs of sequences. For that purpose, we calculated all the pairwise distances between all whole genome sequences, both paired and unpaired. In this case, the most suitable distance model was GTR+G. The regression models between these distances and the variables “time between infections”, “same individual”, and “type” were fitted using the lm() function in R. Two models were fitted, whose calls were lm(dist~time + same_ind) and lm(dist~time + same_ind + type) for models 1 and 2, respectively. The variables were the pairwise distance between sequences (dist), time in days between infections (time), whether the sequences belonged to the same individual (same_ind; 0–no; 1–yes), and the combination of variants for each pairwise comparison of sequences (type). Model 1 was fitted with all the pairwise distances. Model 2 also added the variable lineage, and sequences with a low-represented lineage, which were those other than the B.1.177 lineage, and Alpha, Delta, and Omicron variants of concern (VOC), were removed to avoid loss of statistical power.

### 2.6. Statistical Analyses

Epidemiological data were described using medians and interquartile ranges (IQR) for continuous variables and proportions for categorical variables. To compare the prevalence of comorbidities and GMA between the RECOVID cohort and the overall Catalan population, we employed the Chi-square test, assuming a total Catalan population of 7,896,306 individuals (December 2022). The descriptive statistical analysis was performed with base functions of R 4.3 [20]. The graphs were generated using Python 3.12 [21] (matplotlib [22], seaborn [23], plotly [24]) and R 4.4 (ggpubr [25], ggalluvial [26], ggridges [27]).

## 3. Results

### 3.1. Participant Characteristics

Of the 3303 participants with at least two positive SARS-CoV-2 test results, we selected 2344 participants with a confirmed reinfection. Participants who did not meet these criteria were excluded from further analysis and classified as either extended infections (*n* = 21) or uncertain reinfections (*n* = 938) (Figure 1). Among the 2344 participants selected, only one individual suffered three SARS-CoV-2 infections during the study period, the remainder experiencing only two.

The temporal distribution of all infections revealed distinct peaks corresponding to the infection waves that took place in Catalonia. The first increase in infections occurred in March 2020, followed by subsequent peaks in October 2020, November 2021, and May 2022 (Figure 2a). These peaks represent periods of heightened infection rates, likely associated with the emergence and spread of new variants or the relaxation of public health measures. The median time between the first and second episodes was 364 days, IQR (200–464) (Figure 2b).

The cohort selected consisted of 72.2% females, with a median age of 45 years IQR (28–63) (Figure 2c). The age distribution within the study population revealed different patterns for each sex. For females, three noticeable peaks were observed: the first around age 25, the second around age 50, and the third gradual rise beginning at age 75. In contrast, the distribution among males was flatter, with overall lower density across most age groups and no clearly defined peaks.

Of the cohort, only 421 participants (17.9%) were fully or partially vaccinated before the first infection, while 1 882 participants (80.3%) were vaccinated before their second infection. Most participants (*n* = 2241, 95.6%) were not hospitalized for either of their SARS-CoV-2 infections. Of the 103 participants (4.4%) who were hospitalized due to their infection, 86 individuals (83.5%) required hospitalization during their first infection, while 11 participants were hospitalized during their second infection, and 6 were hospitalized during both infections. Among them, 18 required ICU admission during their first infection and 1 during their second. A full description of the cohort is shown in Table 2.

We next assessed the health status of the participants included. Most reinfected individuals had at least one chronic condition (*n* = 2028, 86.5%), regardless of age, suggesting an overall compromised health status in reinfected individuals (Figure 2d). Notably, a high prevalence of chronic conditions was also observed in younger individuals: 70.9% among those aged 0–17 years and 77.3% in the 18–25 age groups, unexpectedly high rates for these age ranges. Overall, 42.9% of the cohort presented a chronic condition affecting four or more systems, compared to 24.5% of the general population of Catalonia (MSIQ, CatSalut) (Table 3). Some comorbidities were underrepresented in our cohort, such as chronic kidney failure (0.2-fold change, *p* = 0.0001), malignant neoplasm (0.41-fold change, *p* = 0.0001), arthritis (0.14-fold change, *p* = 0.0001), and Chronic Obstructive Pulmonary Disease (0.75-fold change, *p* = 0.0059). Other comorbidities were overrepresented in the RECOVID cohort, including cirrhosis (3.1-fold change, *p* = 0.0001), type I and II diabetes (1.4-fold change, *p* = 0.0001), ischemic heart disease (1.3-fold change, *p* = 0.0097), and high blood pressure (1.2-fold change, *p* = 0.0001). Additionally, 0.4% of the reinfected participants experienced an organ transplant, while only about 0.01% of the Catalan population receives transplants each year (23.5-fold change, *p* = 0.0001).

### 3.2. Impact of Vaccination on SARS-CoV-2 Reinfections

Of all participants, 19.8% experienced SARS-CoV-2 reinfection, having received no vaccination before either the initial or second infection. Among the remaining participants, 80.3% had received at least one dose of the vaccine before their second infection. As illustrated in Figure 3a, we categorized the participants based on their vaccination and reinfection timelines: 4.6% of individuals were vaccinated before their first infection and were reinfected without having received a booster (VII[B] group); 34.9% were vaccinated between infections without a booster (IVI[B] group); 27.2% were both vaccinated and boosted between infections (IVBI group); and 13.5% were vaccinated before their first infection and later boosted between infections (VIBI group). Importantly, approximately half of the participants in the groups vaccinated between infections (IVI[B] and VIBI), with or without a booster, experienced partial breakthrough reinfections. This suggests that the vaccine had not yet reached full efficacy at the time of the second exposure.

The density of days between vaccination and second infection is shown in Figure 3b. We observed a high incidence of reinfections within the first 40 days post-vaccination, followed by a sharp decline between 40 and 100 days, and then a second peak. This trend was consistent regardless of the order of vaccination and initial infection.

Regarding hospitalization in the context of vaccination, 80 of the 86 participants hospitalized during their first infection were not vaccinated (93%), as they were infected during the early waves before vaccines were available. Only one individual requiring ICU admission was vaccinated prior to infection but tested positive for SARS-CoV-2 just 5 days post-vaccination. However, a few individuals were hospitalized during their second infection (*n* = 11, 0.47%) or both infections (*n* = 6, 0.26%), with approximately half of these individuals (*n* = 6, 54.5% and *n* = 3, 50%, respectively) having been vaccinated before the time of reinfection. Only one individual required ICU admission during the second infection and had received only a single dose of the BNT162b2 vaccine.

### 3.3. Impact of SARS-CoV-2 Variants on Reinfection Patterns

As WGS was only performed for a subset of cases, we first inferred the SARS-CoV-2 variant infecting each participant (*n* = 2344) based on the dominant circulating variant in Catalonia at the time of their infection, using GISAID data (www.gisaid.org, Appendix A, accessed on 25 July 2022). The most common reinfection pattern observed was an initial infection with a pre-Alpha lineage followed by an Omicron infection, accounting for nearly 40% of the cases (Appendix A). The next most frequent combination involved pre-Alpha followed by Delta VOC (17%). Cases where both infections involved two pre-Alpha lineages occurred in just over 10% of individuals. In contrast, reinfection events where the first and second infections involved the same VOC—specifically Alpha–Alpha, Delta–Delta, or Omicron–Omicron combinations—were exceptionally uncommon, with frequencies approaching zero.

To perform a more precise genetic analysis of SARS-CoV-2 variants, we analyzed 74 pairs of viral sequences and 52 unpaired sequences using various methods. In the first infection, the most common lineages were pre-Alpha B.1.177 (35.2%) and the Delta variant AY.43 (29.6%). In the second infection, Omicron lineages predominated, with BA.1.1 (20.6%), BA.1.17 (23.5%), and BA.1.1.1 (10.8%) being the most abundant, and some Delta lineages, particularly AY.43 (12.7%). The proportions of all variants found are detailed in Table 4. When the variant pairs were examined, the most common combinations of variants observed between the two infections were Delta–Omicron, followed by B.1.177–Omicron, B.1.177–Delta, and Alpha–Omicron (Figure 4a). Phylogenetic analysis of the 74 sequence pairs revealed substantial genetic divergence between the spike genes of first and second infection variants, particularly between Omicron and earlier strains like pre-Alpha and Delta (Figure 4b). This supports the finding that reinfections typically involve shifts to newer variants, with Omicron emerging as the dominant strain in second infections.

To examine the genetic distances between the sequences, we applied two linear models (Table 5). The first model aimed to analyze the genetic distance between viral sequences from different infections, factoring in two variables: the time between infections and whether the sequences came from the same individual or different individuals (distance~time + same_individual). This model revealed a positive relation between the genetic distance between sequences and the time between infections (estimate = 3.514 × 10^−6^; *p* < 2 × 10^−16^), meaning greater genetic divergence with longer intervals between infections. The model also suggested that viruses infecting the same individual were genetically less related than those infecting different individuals (estimate = 3.243 × 10^−4^; *p* = 0.00198). This result could be influenced by data imbalance, as only 148 pairwise distances were from the same individual, compared to 40,253 from different individuals. The adjusted R-squared of 0.254 indicates that the model explains 25.4% of the genetic distance variation, highlighting the importance of time between infections and individual-specific factors.

The second model (distance~time + same_individual + type) included SARS-CoV-2 lineage/VOC as a variable, focusing on B.1.177, Alpha, Delta, and Omicron. The time variable remained significant, with a positive correlation, while the “same individual” variable lost significance. Regarding variants, all combinations of them, except B.1.177-Delta, showed *p*-values < 2 × 10^−16^. The estimates were positive for different VOC pairs (e.g., Delta-Omicron) and negative for the same VOC pairs (e.g., Delta–Delta). The adjusted R-squared of this model was 0.9537, showing that the primary factor driving genetic distance is the evolution of SARS-CoV-2 over time.

## 4. Discussion

Our study provides new insights into the factors contributing to SARS-CoV-2 reinfection, with a particular emphasis on the role of the host’s health status. We found that a significant proportion of participants had chronic conditions, suggesting a potential link between compromised health and increased susceptibility to reinfection—an aspect not commonly explored in previous research. While the genetic variability of the virus remained the primary driver of reinfection [28,29], our findings also underscore the impact of underlying health conditions as an important factor. Additionally, waning immunity seemed to play a significant role, as the median time between infections was one year [30]. Regarding vaccination, while it was crucial in reducing disease severity, its effectiveness in preventing reinfections was limited. We observed a decrease in reinfection incidence between 40 and 100 days post-vaccination, after which the incidence increased, reflecting a diminishing protective effect.

Although SARS-CoV-2 reinfections are a major problem, the scientific literature assessing this topic remains limited. Some established risk factors for SARS-CoV-2 reinfection include waning immunity, viral evolution, and variations in immune responses among individuals. Moreover, studies have demonstrated that reinfections could be associated with specific variants, such as the Delta and Omicron VOCs [31,32,33], which exhibit key mutations in the spike protein. Other risk factors described for reinfection are being unvaccinated, infrequent face mask use, time since first infection, being non-Hispanic black, a severe first infection, or being over 60 years old [34,35]. Our results suggested that the median time between infections was 1 year, reflecting both loss of immunity over time and the emergence of new SARS-CoV-2 variants [36].

Our temporal data illustrates the distribution of reinfections at three key time points: 200, 400, and 600 days between infections, with a median of 364 days. These distributions were directly associated with the emergence of significant new SARS-CoV-2 VOCs, such as Delta, appearing seven months after the summer of 2020, and the Omicron VOC in January 2022 [37]. Therefore, our data align with the high rates of reinfection involving new variants that other researchers have previously documented [38]. The genetic distance models also hint at the impact of the genetic variability of the virus in the reinfection phenomenon.

Our cohort consisted primarily of women, with the age density curve showing three maxima over different age groups being much more prominent in women. The first two peaks correspond to ages 24 and 45. A feasible explanation for this is the nature of the cohort: all the samples analyzed here were taken from hospitalized or in-and-out patients and healthcare workers, which suggests that the higher number of women in our sample is in part attributable to a significant proportion of them being hospital staff and the prominent caregiving role of women in society [39]. In addition, it has been demonstrated that women tend to prioritize their health more than men and are therefore more likely to schedule medical appointments [40]. The third peak, around 85 years, is arguably filled with women living in long-term care facilities, who generally have a longer life expectancy than men [41]. At this point, no sex/gender-related susceptibility to reinfection has been reported, so we assume the sex bias observed in our cohort is due to the sample’s composition rather than biological factors.

Our AMG data revealed that the RECOVID cohort generally had poor health, with more than 86% of participants having at least one chronic condition. However, the median age of the cohort was 45 years, a distribution similar to that of the overall Catalan population (mean of 43.3 years, data obtained from the Statistical Institute of Catalonia, Idescat, 2022). Thus, age does not appear to be the main factor driving the poor health observed in our cohort. Notably, over 75% of younger individuals (<26 years old) also had one or more chronic conditions. This suggests that the probability of reinfection increases with chronic comorbidities [42]. The increased reinfection rate among individuals with chronic conditions may be attributed to several factors. First, chronic diseases can impair immune function, reducing the ability to develop an effective response to SARS-CoV-2. Second, medications commonly used to manage chronic conditions (i.e., corticosteroids or immunosuppressants) may further weaken immune functions. In addition, it is also possible that individuals with chronic conditions have more frequent contact with healthcare services, increasing their exposure risk and likelihood of testing. While we did not assess viral load in this study, we acknowledge that higher viral loads in immunocompromised individuals could enhance RT-PCR detection sensitivity. Further studies exploring the interplay between viral kinetics, host immunity, and comorbidity burden are warranted.

Although the main goal of vaccination has been to prevent severe illness and hospitalization due to SARS-CoV-2 infection [43], its impact on reinfection has not been widely studied. Previous research has shown a 27–35% reduced risk of reinfection after vaccination [44], although this protective effect is limited. Similarly to this, and in line with the fact that the humoral immune response declines after 3 months of vaccination [45], our data showed a protective effect of the vaccination that lasted between 40 and 100 days after administration, but beyond 100 days, reinfection rates increased. As expected, we observed a higher number of hospitalizations during the first infection, with 93% of hospitalizations occurring among unvaccinated individuals. In contrast, the number of individuals requiring hospitalization during their reinfection was 0.68%. This is attributable to the success of the vaccination campaign in Catalonia, which managed to fully vaccinate 85% of the population by May 2022. Although 15% of the general population remained unvaccinated, 50% of study participants hospitalized during their reinfection were unvaccinated, highlighting the effectiveness of current vaccination regimens in preventing hospitalization [3]. Globally, women have generally experienced milder COVID-19 and lower hospitalization rates compared to men [39], which may have also contributed to the relatively low proportion of hospitalizations observed in our predominantly female cohort.

Our viral genomics results and genetic distance analysis suggest that the main driver for SARS-CoV-2 reinfection is the emergence of new variants over time. However, this could be biased by the definition of reinfection, which requires a gap of more than 90 days between the two infections. Nevertheless, the dominance period of each predominant variant exceeded 90 days (pre-Alpha period: 331 days, Alpha: 140 days, Delta: 175 days, and Omicron: 162 days, until the end of the study), allowing two consecutive infections with the same variant. This finding highlights the global need to implement immunological surveillance in the face of possible genetic modifications of the virus that could evade the immune response generated by existing vaccines, especially in sensitive individuals such as those with comorbidities [46].

This study presents some limitations, primarily due to its retrospective design and lack of control groups of participants with a single infection. The absence of this control group limited our ability to apply certain statistical analyses, such as hypothesis testing or odds ratios. Moreover, as the data was completely anonymized, we were unable to access certain relevant information, such as the characteristics of symptoms during the acute infection period or whether participants developed long COVID. The phylogenetic analysis was limited to 74 reinfection pairs with high-quality sequences, which constrains the statistical power and generalizability of the findings. Furthermore, we were unable to retrieve additional sequences due to many of the samples having a high Ct value, making it impossible to sequence them effectively, which would have strengthened the genomic analysis. Finally, we lacked information about the immune response of the participants in any of the infections, which would have been extremely valuable to understanding the development of these reinfections. Nevertheless, our study has some strong points. First, it involved a relatively large number of participants, mainly due to the fact that SARS-CoV-2 testing was carried out on a routine basis in healthcare facilities and nursing homes, increasing the possibility of detecting reinfections even in asymptomatic individuals. Thus, we were able to extensively characterize the clinical timeline of individual participants. In addition, our findings suggest a potential link between compromised health and increased susceptibility to reinfections and their decrease between 40 and 100 days after vaccination, something to be borne in mind by clinicians working with susceptible patient populations, as well as by those involved in designing health policies.

## 5. Conclusions

Our study sheds light on some of the factors contributing to SARS-CoV-2 reinfection, with genetic variability of the virus being the primary driver, followed by the infected individual’s underlying health status. We also found that vaccination, while essential for reducing disease severity, provided only a short-term protective effect against reinfections lasting around two months. Prospective studies evaluating more risk factors associated with reinfections and their clinical consequences—not least their impact on Long COVID—are needed.

## Figures and Tables

**Figure 1 viruses-17-00840-f001:**
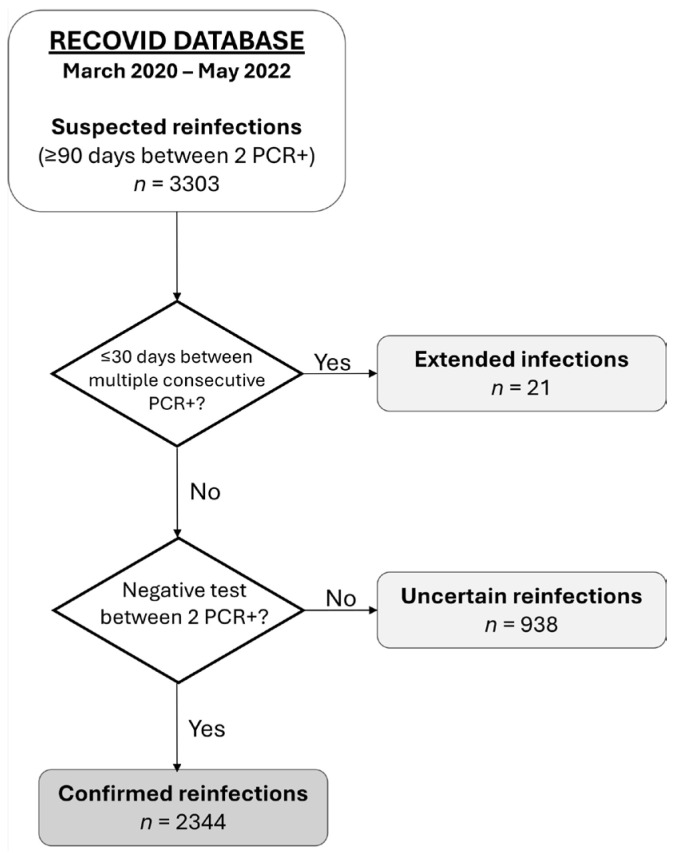
Inclusion flowchart for the RECOVID cohort.

**Figure 2 viruses-17-00840-f002:**
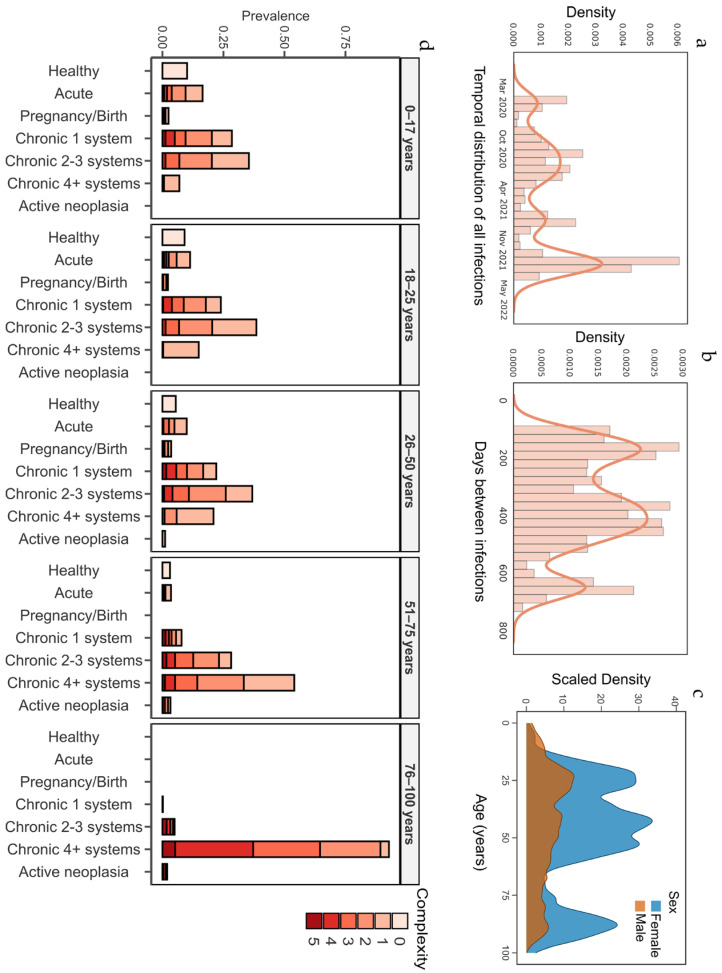
Cohort demographics. (**a**) Temporal distribution of all infections. (**b**) Distribution of number of days between episodes. (**c**) Age distribution of RECOVID cohort by sex. (**d**) AMG in different age groups (far-left: 0–17, left-center: 18–25 years; center: 26–50 years; right-center: 51–75 years; far-right: 76–100 years). The intensity of red shading increases with the healthcare complexity of the participants.

**Figure 3 viruses-17-00840-f003:**
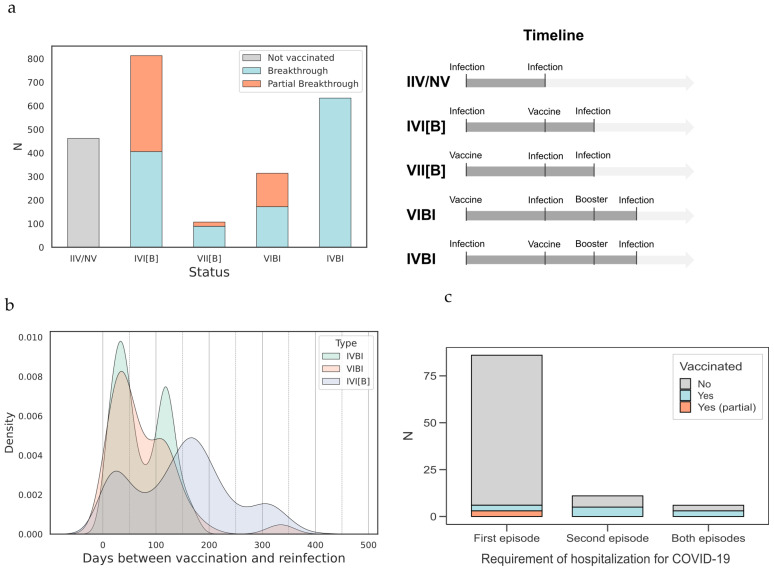
Vaccination data. (**a**) Number of participants broken down by vaccination and infection chronology. Group labels are defined as: NV—not vaccinated; I—infection; V—vaccine; B—booster dose; [B]—possible booster dose (not relevant); the timeline of the events for each group is illustrated in the right panel. The color coding indicates vaccination status: Gray—not vaccinated/vaccinated after both infections. Blue—breakthrough infection. Orange—partial breakthrough infection. (**b**) Density of days between vaccination and reinfection. The days are calculated using the dates of the first dose and the first positive PCR of the second infection. The three colors depict the different types of infection-vaccine chronology. (**c**) Number of COVID-19-related hospitalized participants and vaccination status. The x-axis shows the COVID-19 episode where hospitalization was required. Green—not vaccinated before hospitalization. Orange—vaccinated before hospitalization. Blue—partially vaccinated before hospitalization.

**Figure 4 viruses-17-00840-f004:**
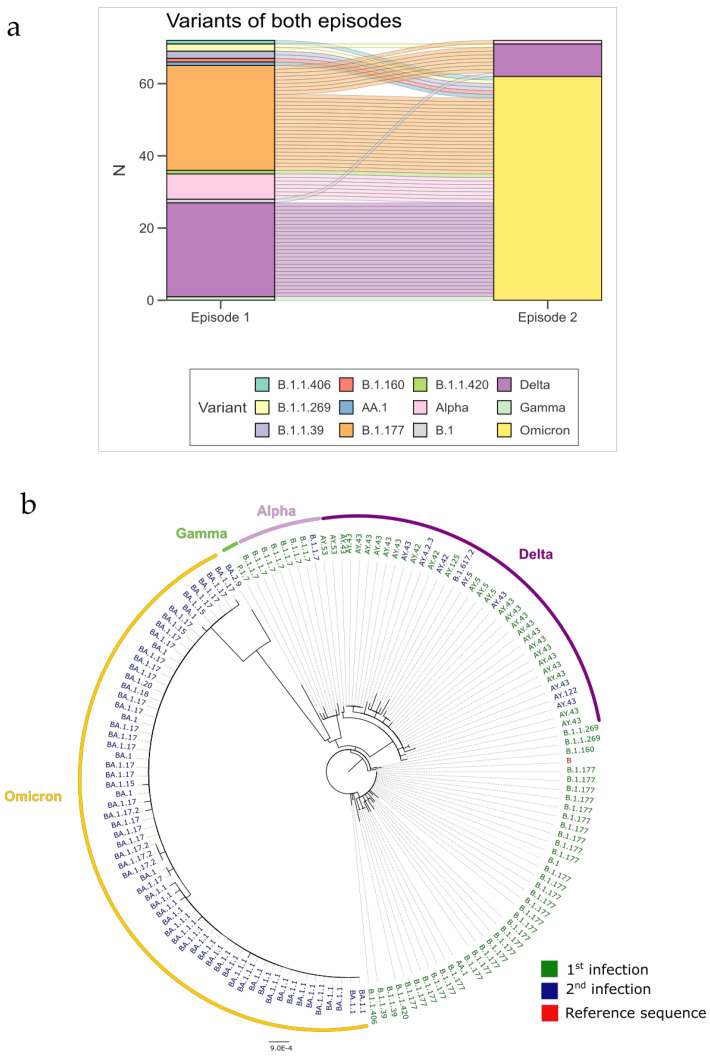
Genomic data. (**a**) Sankey plot of the change in variant between episodes of 74 participants. The sub-variants belonging to Alpha, Delta, and Omicron have been grouped for representational purposes. (**b**) Phylogenetic tree of the spike sequence of the sequences of both episodes of 74 patients. The labels show the lineage, and the colors depict the episode of the infection (green—first infection; blue—second infection; red—reference sequence).

**Table 1 viruses-17-00840-t001:** Definitions.

Concept	Definition
Reinfection	Reinfection occurs when a person contracts SARS-CoV-2 infection, recovers, and becomes infected again. In our cohort, we define reinfection as two positive PCR tests separated by at least 90 days, with a negative test in between.
Extended infection	An extended infection is a single infection episode, lasting longer than usual, defined by at least two positive PCR tests separated by less than 30 days. Participants with extended infection were identified by reviewing complex cases in our cohort.
Complete vaccination schedule	A complete vaccination schedule was defined as receiving two doses of BNT162b2 (Pfizer-BioNTech, Mainz, Germany), mRNA-1273 (Moderna, Cambridge, MA, USA), and/or AZD1222 (AstraZeneca, Cambridge, United Kingdom), or a single dose of Ad26.COV2.S (Janssen, Leiden, The Netherlands) vaccine, with at least 14 days elapsed since the final dose.
Breakthrough infection	A breakthrough infection occurs when a fully vaccinated individual contracts SARS-CoV-2 at least two weeks after the final dose (first dose of viral vector vaccine or second dose of mRNA vaccine).
Partial breakthrough infection	A partial breakthrough infection occurs when a partially vaccinated individual contracts SARS-CoV-2, either after a single dose of an mRNA vaccine or less than two weeks after the final dose (first dose of viral vector vaccine or second dose of mRNA vaccine).
Booster dose	A booster dose is an additional vaccine dose after the primary vaccination series to enhance or prolong immunity against SARS-CoV-2. We also regard as a booster dose any dose given after a breakthrough or partial breakthrough infection.
Adjusted Morbidity Groups (AMG) [10]	Adjusted Morbidity Groups (AMG) [10] is a measure of a person’s general health status, developed by the Catalan Health Institute. It is based on two indicators: group and complexity. The group categories include “healthy”, “acute disease [of any type]”, “pregnant/giving birth”, “chronic condition in one system”, “chronic condition in two or three systems”, “chronic condition in four or more systems”, and “active neoplasia”. Complexity ranges from 0 to 5, reflecting an individual’s personal healthcare needs.

**Table 2 viruses-17-00840-t002:** Descriptive data for the RECOVID cohort (*n* = 2344).

	RECOVID
Number of confirmed reinfections (2 episodes), *n*	2343
Number of confirmed reinfections (3 episodes), *n*	1
Sex, females, *n* (%)	1692 (72.2)
Age in years, median (IQR)	45 (28–63)
Time between infections in days, median (IQR)	364 (200–464)
Hospitalization related to COVID-19, *n* (%)	103 (4.4)
Hospitalization—first infection, *n* (%)	86 (3.6)
Hospitalization—second infection, *n* (%)	11 (0.5)
Hospitalization—both infections, *n* (%)	6 (0.3)
Vaccinated before first infection ^1^, *n* (%)	421 (18)
Vaccinated before second infection ^1^, *n* (%)	1882 (80.3)

^1^ Vaccination status includes both fully and partially vaccinated individuals.

**Table 3 viruses-17-00840-t003:** Adjusted morbidity groups and comorbidities of the RECOVID cohort compared with the general population of Catalonia.

	Prevalence (%)		Chi-Square *p*-Value ^2^
	RECOVID Cohort	Catalonia Population ^1^	
AMG			
Healthy	4	17.8	*0.0001*
Acute disease (any)	7.2	8	*0.1586*
Pregnancy/birth	2.5	1.3	*0.0001*
Chronic disease 1 system	14	18.7	*0.0001*
Chronic disease 2–3 systems	28	25.3	*0.0028*
Chronic disease 4+ systems	42.9	24.5	*0.0001*
Active neoplasm (any)	1.6	4.4	*0.0001*
Comorbidities			
Diabetes (Type I and II)	11.2	8.2	*0.0001*
COPD ^3^	3.6	4.8	*0.0059*
Asthma	6.8	7	*0.6809*
Ischemic heart disease	3.8	2.9	*0.0097*
Chronic heart failure	3.5	3.4	*0.7929*
High blood pressure	23.6	20.4	*0.0001*
Chronic kidney failure	0.97	4.8	*0.0001*
Cirrhosis	1.1	0.35	*0.0001*
HIV ^4^ infection	0.24	0.42	*0.2195*
Malignant neoplasm (any)	2.9	7	*0.0001*
Arthritis	0.9	6.3	*0.0001*
Organ transplant (any)	0.4	0.017	*0.0001*

^1^ Data regarding the Catalan population was obtained from the Catalan Health Institute (CatSalut, 2021) and the Catalan Transplant Organization (OCATT, 2021). ^2^ For Chi-square calculations, we assumed a total Catalan population of 7,896,306 individuals. ^3^ COPD: Chronic obstructive pulmonary disease. ^4^ HIV: Human Immunodefiency virus

**Table 4 viruses-17-00840-t004:** SARS-CoV-2 lineages of sequenced samples.

Lineage	Variant	Episode 1 N (%)	Episode 2 N (%)
AA.1	-	1 (1.1)	0 (0)
AY.122	Delta	0 (0)	1 (0.9)
AY.125		1 (1.1)	1 (0.9)
AY.127		0 (0)	1 (0.9)
AY.36		0 (0)	1 (0.9)
AY.4		0 (0)	1 (0.9)
AY.4.2.3		0 (0)	1 (0.9)
AY.42		2 (2.1)	1 (0.9)
AY.43		26 (27.7)	13 (12.3)
AY.5		3 (3.2)	1 (0.9)
AY.5.4		0 (0)	1 (0.9)
AY.53		2 (2.1)	0 (0)
AY.6		1 (1.1)	0 (0)
AY.71		0 (0)	1 (0.9)
B.1	-	1 (1.1)	0 (0)
B.1.1.269	-	3 (3.2)	0 (0)
B.1.1.39	-	2 (2.1)	0 (0)
B.1.1.406	-	1 (1.1)	0 (0)
B.1.1.420	-	1 (1.1)	0 (0)
B.1.1.7	Alpha	11 (11.7)	2 (1.9)
B.1.160	-	1 (1.1)	0 (0)
B.1.617.2	-	0 (0)	1 (0.9)
B.1.1.177	-	36 (38.3)	0 (0)
BA.1	Omicron	0 (0)	9 (8.5)
BA.1.1		0 (0)	22 (20.8)
BA.1.1.1		0 (0)	11 (10.4)
BA.1.15		0 (0)	4 (3.8)
BA.1.17		1 (1.1)	24 (22.6)
BA.1.17.2		0 (0)	5 (4.7)
BA.1.18		0 (0)	1 (0.9)
BA.1.20		0 (0)	1 (0.9)
BA.2		0 (0)	1 (0.9)
BA.2.9		0 (0)	2 (1.9)
P.1.7	Gamma	1 (1.1)	0 (0)
Total		94	106

**Table 5 viruses-17-00840-t005:** Linear models of genetic distance.

Model 1				
Coefficients				
	Estimate	Std. error	*p*-value	
(Intercept)	1.15 × 10^−3^	1.03 × 10^−5^	<2 × 10^−16^	
Time	3.51 × 10^−6^	4.26 × 10^−8^	<2 × 10^−16^	
Same individual	3.24 × 10^−4^	1.05 × 10^−4^	0.00198	

Adjusted R-squared	0.2539			
F-statistic *p*-value	<2.2 × 10^−16^			
Model 2				
Coefficients				
	Estimate	Std. error	*p*-value	
(Intercept)	1.59 × 10^−3^	1.12 × 10^−5^	<2 × 10^−16^	
Time	1.54 × 10^−7^	2.9 × 10^−8^	1.98E-07	
Type Alpha–Delta	6.57 × 10^−4^	1.30 × 10^−5^	<2 × 10^−16^	
Type Alpha–Omicron	1.01 × 10^−3^	1.29 × 10^−5^	<2 × 10^−16^	
Type B.1.177–B.1.177	−1.17 × 10^−3^	1.45 × 10^−5^	<2 × 10^−16^	
Type B.1.177–Delta	9.38 × 10^−6^	1.20 × 10^−5^	0.436	
Type B.1.177–Omicron	6.99 × 10^−4^	1.39 × 10^−5^	<2 × 10^−16^	
Type Delta–Delta	−1.15 × 10^−3^	1.22 × 10^−5^	<2 × 10^−16^	
Type Delta–Omicron	1.39 × 10^−3^	1.09 × 10^−5^	<2 × 10^−16^	
Type Omicron–Omicron	−1.25 × 10^−3^	1.16 × 10^−5^	<2 × 10^−16^	
Same individual	−1.44 × 10^−5^	2.93 × 10^−5^	0.623	

Adjusted R-squared	0.9537			
F-statistic *p*-value	<2.2 × 10^−16^			

## Data Availability

Viral genomic data generated were uploaded to the GISAID database (www.gisaid.org) and can be accessed through the set ID EPI_SET_250311pz.

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
