# Peer review of "Identification of Clinical and Genomic Features Associated with SARS-CoV-2 Reinfections"

_viruses, 2025, doi:10.3390/v17060840_

Round 1

Reviewer 1 Report

Comments and Suggestions for Authors

In the manuscript, the authors attempted to establish a relationship between the frequency (duration between infwections) of reinfection and the severity of (re)infection and the absence or presence of vaccination, health status, sex and age of patients, and variability of the virus. Although all the results obtained and conclusions drew are obvious and were already known, they are valuable as a regional demonstration of their reliability (in this case, using the example of Catalonia). The manuscript may be published in Viruses MDPI after taking into account the reviewer's comments:

  1. Reference 3: provide article number and DOI.

References 4,5: provide article number

Reference 6: provide DOI

Reference 7 has already published: https://doi.org/10.1093/ofid/ofad493

And so on.

Please, check all the references

  1. Abstract:

In this study, we examined data for 3 303 27 individuals who had had two SARS-CoV-2 infections between March 2020 and May 2022 from both clinical and viral genomics perspectives.

«had» appears twice. The same for line 227

  1. Line 57: «Long COVID» should be «long COVID»

Line 65-73: «The introduction should briefly place the study in a broad context and highlight why it is  important. It should define the purpose of the work and its significance. The current state of the research field should be carefully reviewed and key publications cited. Please highlight controversial and diverging hypotheses when necessary. Finally, briefly mention the  main aim of the work and highlight the principal conclusions. As far as possible, please keep the introduction comprehensible to scientists outside your particular field of research. References should be numbered in order of appearance and indicated by a numeral or numerals in square brackets—e.g., [1] or [2,3], or [4–6]. See the end of the document for further details on references.»

Delete this guideline for authors

  1. Line 80 and 88: HUGTiP and HUGTIP, use any one of them
  2. Line 134-135: should be «Genomic reverse transcription, tiling PCR amplification, and sequencing were performed as previously described [10].»
  3. Line 139-141 should be «The sequences were analyzed with the nf-core/viralrecon pipeline [11,12] which executes all the steps, from quality control and alignment of raw sequences to SARS-CoV-2 140 lineage assignment using Nextclade [13] and Pangolin [14].»
  4. Line 175-177: «This section may be divided by subheadings. It should provide a concise and precise description of the experimental results, their interpretation, as well as the experimental conclusions that can be drawn.» should be deleted.
  5. Line 192-194 and so on: The median time between first and second episodes was 364 days [200–464] (Figure 2B). The cohort selected consisted of 72.2% females, with a median age of 45 years [28-63] (Figure 2C).

I suggest no to use square brackets because they look like brackets for references.

  1. Line 196-199: «For females, three noticeable peaks were observed: the first around age 25, the second around age 50, and the third around age 75. In contrast, the male population showed a significantly lower density across nearly all age groups, with a less variable distribution and a prominent peak around age 50.» Please, add Figure 2C at the end.

For females, the third began at age 75. The reviewer did not see a prominent peak around age 50 for males.

  1. Line 203: «(top left: 0–25 years; top right: 26-50 years; bottom left: 51–75 years; bottom right: 76–100 years).» They are all in one line.
  2. Line 254-255: «Grey – Not vaccinated/Vaccinated after both infections; Blue – Breakthrough infection; Orange – Partial breakthrough infection.»

This information is presented in Figure 3A and is therefore redundant.

Author Response

Reviewer 1                     

In the manuscript, the authors attempted to establish a relationship between the frequency (duration between infections) of reinfection and the severity of (re)infection and the absence or presence of vaccination, health status, sex and age of patients, and variability of the virus. Although all the results obtained and conclusions drew are obvious and were already known, they are valuable as a regional demonstration of their reliability (in this case, using the example of Catalonia). The manuscript may be published in Viruses MDPI after taking into account the reviewer's comments:

  1. Reference 3: provide article number and DOI. References 4,5: provide article number; Reference 6: provide DOI; Reference 7 has already published: https://doi.org/10.1093/ofid/ofad493; And so on. Please, check all the references

We thank the reviewer for pointing out this omission. We have carefully reviewed and revised all references, and updated them to include the article numbers and DOI identifiers where available. Reference 7 has also been updated with the final published version and its DOI.

  1. In this study, we examined data for 3 303 27 individuals who had had two SARS-CoV-2 infections between March 2020 and May 2022 from both clinical and viral genomics perspectives. «had» appears twice. The same for line 227

We thank the reviewer for their attention to detail. The use of "had had" in the sentence is grammatically correct, as it employs the past perfect tense to indicate that the individuals had experienced two SARS-CoV-2 infections prior to the time of data analysis. However, to improve clarity and avoid potential confusion, we have rephrased both sentence as follows:

Line 30 of the new version: "In this study, we examined data for 3,303 individuals who experienced two SARS-CoV-2 infections between March 2020 and May 2022..."

Line 241: “Additionally, 0.4% of the reinfected participants experienced an organ transplant while only about 0.01% of the Catalan population receive transplants each year (23.5-fold change, p=0.0001).”

  1. Line 57: «Long COVID» should be «long COVID»

We have modified Long COVID for long COVID throughout the text.

  1. Line 65-73: «The introduction should briefly place the study in a broad context and highlight why it is important. It should define the purpose of the work and its significance. The current state of the research field should be carefully reviewed and key publications cited. Please highlight controversial and diverging hypotheses when necessary. Finally, briefly mention the main aim of the work and highlight the principal conclusions. As far as possible, please keep the introduction comprehensible to scientists outside your particular field of research. References should be numbered in order of appearance and indicated by a numeral or numerals in square brackets—e.g., [1] or [2,3], or [4–6]. See the end of the document for further details on references.» Delete this guideline for authors

We apologize to the reviewer for the oversight. The journal's template guideline mistakenly remained in the manuscript, and we have now removed this section.

  1. Line 80 and 88: HUGTiP and HUGTIP, use any one of them

We have unified the name of our hospital to HUGTiP.

  1. Line 134-135: should be «Genomic reverse transcription, tiling PCR amplification, and sequencing were performed as previously described [10].»

We have modified the sentence.

  1. Line 139-141 should be «The sequences were analyzed with the nf-core/viralrecon pipeline [11,12] which executes all the steps, from quality control and alignment of raw sequences to SARS-CoV-2 140 lineage assignment using Nextclade [13] and Pangolin [14].»

We have modified the sentence.

  1. Line 175-177: «This section may be divided by subheadings. It should provide a concise and precise description of the experimental results, their interpretation, as well as the experimental conclusions that can be drawn.» should be deleted.

We apologize to the reviewer for the oversight. The journal's template guideline mistakenly remained in the manuscript, and we have now removed this section.

  1. Line 192-194 and so on: “The median time between first and second episodes was 364 days [200–464] (Figure 2B). The cohort selected consisted of 72.2% females, with a median age of 45 years [28-63] (Figure 2C)”. I suggest no to use square brackets because they look like brackets for references.

As reviewer suggested, we have modified all median [IQR] for median (IQR) to avoid confusion with the references.

  1. Line 196-199: «For females, three noticeable peaks were observed: the first around age 25, the second around age 50, and the third around age 75. In contrast, the male population showed a significantly lower density across nearly all age groups, with a less variable distribution and a prominent peak around age 50.» Please, add Figure 2C at the end. For females, the third began at age 75. The reviewer did not see a prominent peak around age 50 for males.

We thank the reviewer for the detailed observation regarding Figure 2C. We agree that in females, the third increase in density begins around age 75 rather than forming a distinct peak. This has now been clarified in the revised manuscript text. Regarding the male distribution, we acknowledge that the density is relatively uniform with no clearly defined peaks. Accordingly, we have removed the reference to a peak around age 50 in males and have updated the manuscript to more accurately reflect the observed data patterns.

Line 206-208: “The age distribution within the study population revealed different patterns for each sex. For females, three noticeable peaks were observed: the first around age 25, the second around age 50, and the third gradual rise beginning at age 75. In contrast, the distribution among males was flatter, with overall lower density across most age groups and no clearly defined peaks”.

  1. Line 203: «(top left: 0–25 years; top right: 26-50 years; bottom left: 51–75 years; bottom right: 76–100 years).» They are all in one line.

We thank the reviewer for pointing this out, and we have modified the description in the figure 2C legend accordingly (lines 212-213).

  1. Line 254-255: «Grey – Not vaccinated/Vaccinated after both infections; Blue – Breakthrough infection; Orange – Partial breakthrough infection.» This information is presented in Figure 3A and is therefore redundant.

We thank the reviewer for this helpful comment. While the vaccination status color coding is presented in the left panel of Figure 3A, we have chosen to retain the right panel (timeline schematic) because it serves a different and complementary purpose. Specifically, the colors in the left panel are used primarily to differentiate between breakthrough (blue) and partial breakthrough (orange) infections, as well as non-vaccinated individuals (grey). These categories cut across all infection/vaccination groups and are critical for interpreting the data shown. The right panel, in contrast, illustrates the temporal sequence of vaccination, infection, and booster events for each group (e.g., IIV/NV, VIBI, IVBI), which helps clarify group definitions that might not be intuitive for readers. To further improve clarity, we have updated the figure caption to explicitly state the distinct roles of each panel and the meaning of the color scheme (Lines 266-269)-

Reviewer 2 Report

Comments and Suggestions for Authors

This manuscript is a retrospective observational study that investigates clinical and genomic factors associated with SARS-CoV-2 reinfections in Badalona, Spain, between March 2020 and December 2022, with large scale samples. The manuscript is comprehensive and methodologically sound. The outcomes are emphasised in genomic analysis. However, the baseline and clinical characteristics, and outcome variables require clarification and expansion to enhance the utility and interpretability of the findings.

Comments.
1. Study registration:
While this is not an interventional trial, many longitudinal or cohort studies involving human biological samples are eligible for prospective registration in platforms such as ClinicalTrials.gov or the EU Clinical Trials Register.

Suggest clarifying whether this study was registered in a public registry. If so, include the registry name and registration number. 
If not, this point is omitted.

2. Clarification of PCR target genes (Lines 82–87):
The manuscript describes the use of RT-qPCR assays for SARS-CoV-2 detection but does not specify which viral genes were targeted (e.g., N, ORF1ab, RdRp).
Even if different kits were used, please specify the most common target genes for clarity and reproducibility. This point is important for readers from the molecular diagnostics field.

3. Ambiguity of the term “PI-21-235” (Lines 77 and 97):
The identifier “PI-21-235” appears both in line 77 and line 97. It appears to be the ethics committee approval number in line 97, but its mention in line 77 could confuse readers (e.g. Grant number, Research code number...).

Consider clarifying or removing the first instance (line 77) unless it refers to a study-specific code or grant number. 
If it is the same as the IRB approval, use it only once and in the correct context.

4. Unclear use of “V” notation (Lines 121–125 and Figure 3):
In lines 121-125 and Figure 3, the term "V" is unclear because the primary series comprises two shots (except, Sputnik "Light" and Jcovden; consider one shot as the primary series). Infection following partial primary series (one dose) and full primary series (two doses) may have different characteristics due to vaccine induced immunity.. Furthermore, Table 1 mentions “Partial breakthrough infection,” which does not align with the undefined "V" term.

Consider clearly define the term “V” in the text. 
If possible, you may consider stratifying breakthrough infections into “partial” (after one dose) and “complete” (after the primary series) categories. (For example, using notation such as V₁ and V₂ for clarity and consistency).

5. Booster dose classification:
Did some participants receive more than one booster dose during the study period?

If data are available, consider stratifying booster vaccination status by dose count (e.g., B₁ for one booster, B₂ for two boosters). This may offer additional insight into the impact of booster regimens on reinfection.

6. Age group classification:
The age stratification used (0–25, 26–50, 51-75, 76-100) is potentially problematic. 
The 0–25 group includes both children/adolescents and young adults, who differ substantially in terms of immune responses, behaviour, vaccination eligibility, and exposure settings (e.g., school vs. workplace).

Please justify the rationale for this grouping, or consider subdividing this group (e.g., 0–17 and 18–25) to improve biological and epidemiological relevance. 
Interpretation of reinfection and vaccination effects in this mixed group should be discussed cautiously.

7. Superficial hospitalisation outcomes:
Hospitalisation is reported but not further stratified by clinical severity (e.g., pneumonia, ICU admission, mechanical ventilation required).

If possible, provide more granular data on disease severity during the first infection and reinfection. 
Also consider stratifying outcomes by vaccination status (unvaccinated, partial, full, booster), viral variants, and time since most recent vaccination.

8. Limited genomic pairing (n = 74):
While the sequencing analysis is valuable, but only 74 paired reinfection cases had genomic data available. This issue is acknowledging to the limitations.

Author Response

Reviewer 3

This manuscript is a retrospective observational study that investigates clinical and genomic factors associated with SARS-CoV-2 reinfections in Badalona, Spain, between March 2020 and December 2022, with large scale samples. The manuscript is comprehensive and methodologically sound. The outcomes are emphasised in genomic analysis. However, the baseline and clinical characteristics, and outcome variables require clarification and expansion to enhance the utility and interpretability of the findings.

Comments.

  1. Study registration:

While this is not an interventional trial, many longitudinal or cohort studies involving human biological samples are eligible for prospective registration in platforms such as ClinicalTrials.gov or the EU Clinical Trials Register. Suggest clarifying whether this study was registered in a public registry. If so, include the registry name and registration number.

If not, this point is omitted.

This study was conducted as a retrospective observational analysis using anonymized clinical and genomic data collected as part of routine healthcare procedures. As such, it was not registered in a clinical trial registry, since it does not meet the criteria for prospective trial or interventional study registration.

  1. Clarification of PCR target genes (Lines 82–87):

The manuscript describes the use of RT-qPCR assays for SARS-CoV-2 detection but does not specify which viral genes were targeted (e.g., N, ORF1ab, RdRp).

Even if different kits were used, please specify the most common target genes for clarity and reproducibility. This point is important for readers from the molecular diagnostics field.

We have included the targeted gene of all tests in the following paragraph (lines 80-82): “All samples underwent routine nucleic acid amplification testing (NAAT), including reverse transcription real-time PCR, RT-PCR (Allplex 2019-nCoV assay [RdRp, E, and N genes, Seegene], Alinity [RdRp and N genes, Abbott], Simplexa [ORF1ab and/or S genes, DiaSorin], and GeneXpert [RdRp, E, and N genes, Cepheid]) using CFX96 instruments (Bio-Rad Laboratories) or transcription-mediated amplification (TMA) using the Panther System (ORF1ab gene, Hologic or Procleix, Grífols S.A., Sant Cugat del Vallès, Spain) to detect the presence of SARS-CoV-2 infection.

  1. Ambiguity of the term “PI-21-235” (Lines 77 and 97):

The identifier “PI-21-235” appears both in line 77 and line 97. It appears to be the ethics committee approval number in line 97, but its mention in line 77 could confuse readers (e.g. Grant number, Research code number...). Consider clarifying or removing the first instance (line 77) unless it refers to a study-specific code or grant number.

If it is the same as the IRB approval, use it only once and in the correct context.

We confirm that "PI-21-235" refers to the ethics committee (IRB) approval number. To avoid ambiguity, we will remove its first mention (Line 73 in the new version of the manuscript) and retain it only its second occurrence (Line 95 in the new version of the manuscript), where it is clearly presented as the IRB reference.

  1. Unclear use of “V” notation (Lines 121–125 and Figure 3):

In lines 121-125 and Figure 3, the term "V" is unclear because the primary series comprises two shots (except, Sputnik "Light" and Jcovden; consider one shot as the primary series). Infection following partial primary series (one dose) and full primary series (two doses) may have different characteristics due to vaccine induced immunity.. Furthermore, Table 1 mentions “Partial breakthrough infection,” which does not align with the undefined "V" term. Consider clearly define the term “V” in the text.

If possible, you may consider stratifying breakthrough infections into “partial” (after one dose) and “complete” (after the primary series) categories. (For example, using notation such as V₁ and V₂ for clarity and consistency).

We thank the reviewer for noticing that the term was not clear enough. We have added the following definition regarding the V term (lines 123-124):

In order to study the association between vaccination and reinfection, participants were categorized into four distinct groups based on the timing of their infections (I=infection) and vaccinations (V=first vaccine dose, in any possible case {Janssen dose, Moderna/AZ/Pfizer first dose or first + second doses};, B=booster dose), as follows: 1) VII[B], consisting of individuals vaccinated before their first infection without receiving a booster; 2) IVI[B], individuals vaccinated between infections but without a booster; 3) IVBI, individuals vaccinated and boosted between infections; and 4) VIBI, individuals vaccinated before the first infection and subsequently boosted before the second infection.

Additionally, we have added a definition of a complete vaccination schedule to the definitions table (Table 1).

Complete vaccination schedule

A complete vaccination schedule was defined as received two doses of Pfizer-BioNTech, Moderna and/or AstraZeneca, or a single dose of Janssen vaccine, with at least 14 days elapsed since the final dose.

  1. Booster dose classification: Did some participants receive more than one booster dose during the study period? If data are available, consider stratifying booster vaccination status by dose count (e.g., B₁ for one booster, B₂ for two boosters). This may offer additional insight into the impact of booster regimens on reinfection.

We thank the reviewer for this relevant comment and suggestion. We reviewed the original data, but only one individual received two booster doses during the study period. In this case, the second booster dose was administered after second infection, and it was the last event of the clinical timeline code of the participant.

We cannot study the effects of additional booster doses in our cohort. To clarify this, we have added the following sentence in the methods section (Line 129): “Except for one individual, all participants received a single booster dose.”

  1. Age group classification:

The age stratification used (0–25, 26–50, 51-75, 76-100) is potentially problematic.

The 0–25 group includes both children/adolescents and young adults, who differ substantially in terms of immune responses, behaviour, vaccination eligibility, and exposure settings (e.g., school vs. workplace). Please justify the rationale for this grouping, or consider subdividing this group (e.g., 0–17 and 18–25) to improve biological and epidemiological relevance.  Interpretation of reinfection and vaccination effects in this mixed group should be discussed cautiously.

We agree with the reviewer. We subdivided the 0-25 years age group into 0-17 and 18-25 years as suggested. Interestingly, the results are quite similar in both groups. We have modified the figure 2D including five age groups. We also included this new information in the manuscript (lines 230-233): “Notably, a high prevalence of chronic conditions was also observed in younger individuals: 70.9% among those aged 0–17 years and 77.3% in the 18–25 age groups, unexpectedly high rates for these age ranges.”

  1. Superficial hospitalisation outcomes:

Hospitalisation is reported but not further stratified by clinical severity (e.g., pneumonia, ICU admission, mechanical ventilation required).

If possible, provide more granular data on disease severity during the first infection and reinfection. Also consider stratifying outcomes by vaccination status (unvaccinated, partial, full, booster), viral variants, and time since most recent vaccination.

We thank the reviewer for this insightful comment. We have expanded our reporting of disease severity by including ICU admission data, which was the only consistent and systematically recorded severity outcome available across both infections. Eighteen individuals required ICU care during their first infection, and one during the second. These data have been added to the Results section.

Unfortunately, more detailed severity metrics (such as pneumonia diagnosis, oxygen therapy, or mechanical ventilation) were not consistently captured in our dataset due to the retrospective nature of the study and variability in hospital record documentation during the pandemic surges. As such, we are unable to reliably stratify by these clinical parameters.

Regarding the impact of vaccination status, we now provide data on the proportion of hospitalized individuals who had been vaccinated prior to reinfection, including information on those who had received a booster. We also indicate that nearly all ICU admissions occurred in unvaccinated individuals or shortly after partial vaccination, suggesting limited protective immunity at the time of infection.

With respect to variant-specific analysis, the limited number of hospitalization events did not allow for meaningful stratification by infecting variant or time since vaccination. Moreover, the high vaccination coverage in the general population and the dynamic shift in dominant SARS-CoV-2 variants during the study period introduce confounding factors that limit the ability to isolate the effect of any single variant or vaccination timing on clinical outcomes.

We have added the following lines to the manuscript

Hospitalization section Line 222-223:

“Of the 103 participants (4.4%) who were hospitalized due to their infection, 86 individuals (83.5%) required hospitalization during their first infection, while 11 participants were hospitalized during their second infection, and 6 were hospitalized during both infections. Among them, 18 required ICU admission during their first infection, and 1 during their second.”

Hospitalization in the context of vaccination section Line 279-281 and 284-285:

“Only one individual requiring ICU admission was vaccinated prior infection, but tested positive for SARS-CoV-2 just 5 days post-vaccination. However, a few individuals were hospitalized during their second infection (n=11, 0.47%) or both infections (n=6, 0.26%), with approximately half of these individuals (n=6, 54.5% and N=3, 50%, respectively) having been vaccinated before the time of reinfection. Only one individual required ICU admission during the second infection and had received only a single dose of the Pfizer vaccine.”

  1. Limited genomic pairing (n = 74):

While the sequencing analysis is valuable, but only 74 paired reinfection cases had genomic data available. This issue is acknowledging to the limitations.

We agree with this limitation. We have included it in the Limitations section (Lines 442-443):

“The phylogenetic analysis was limited to 74 reinfection pairs with high-quality sequences, which constrains the statistical power and generalizability of the findings

Reviewer 3 Report

Comments and Suggestions for Authors

 Muñoz-López et al studied the clinical and genomic features of SARS-CoV-2 virus to find the cause of reinfection. They observed that genomic evolution is the main factor behind reinfection.   They also found that chronic infection were common for most reinfection. Based on their results, they concluded more precaution should be taken for vaccination among people with chronic conditions.

They identified important factors for reinfections which are useful. However, the paper has some concerns to be addressed. However, there are some concerns listed here that need to be addressed.

  1. Line 65-73, the templet text remained and amalgamated with introductory text.
  2. Text within supplementary figure 1A and Figure 2b (X-axis label) is illegible and font should be increased.
  3. The text in Figure 3a is illegible especially (timeline) and should be improved.
  4. Authors used phylogenetic analysis using spike protein instead of using whole genome. Authors should write the rationale behind this selection.
  5. Line 358-365, authors should also mention with reference that in global pandemic, women are generally experienced less hospitalization or mildly infection than men for which they observed very less percentage of hospitalization in their cohort.
  6. Authors should provide a separate paragraph explain potential reason for why chronic conditions are mostly reinfected? Is it for less immunity? But chronic conditions generally lead to more hospitalization and morbidity, but an increase in infection has not been documented. It is also possible that medication used for chronic conditions weakens the immunity. Did authors look into the viral load between chronic condition and normal infection? Assuming higher viral load in chronic condition (because they are more vulnerable and hospitalized/morbidities) may lead to higher detection rate in RT-QPCR than normal individuals which could have lower viral loads.

Author Response

Reviewer 2

Muñoz-López et al studied the clinical and genomic features of SARS-CoV-2 virus to find the cause of reinfection. They observed that genomic evolution is the main factor behind reinfection.   They also found that chronic infection were common for most reinfection. Based on their results, they concluded more precaution should be taken for vaccination among people with chronic conditions.

They identified important factors for reinfections which are useful. However, the paper has some concerns to be addressed. However, there are some concerns listed here that need to be addressed.

  1. Line 65-73, the templet text remained and amalgamated with introductory text.

We apologize to the reviewer for the oversight. The journal's template guideline mistakenly remained in the manuscript, and we have now removed this section

  1. Text within supplementary figure 1A and Figure 2b (X-axis label) is illegible and font should be increased.

The font size in Supplementary Figure 1A and the X-axis label in Figure 2B has been increased to improve legibility. We have revised the figures accordingly and ensured that all text is now clear and readable.

  1. The text in Figure 3a is illegible especially (timeline) and should be improved

The font size in Figure 3a and has been increased to improve legibility.

  1. Authors used phylogenetic analysis using spike protein instead of using whole genome. Authors should write the rationale behind this selection.

We used the spike gene for phylogenetic analysis because the spike protein is the primary target of neutralizing antibodies and vaccines, and it accumulates most of the mutations that influence viral infectivity and immune escape. Therefore, it provides critical insights into the antigenic evolution of SARS-CoV-2, which is directly relevant to reinfection risk. Additionally, sequencing coverage in non-spike regions was limited in some samples, with multiple undetermined bases (Ns) in the FASTA files. Restricting the analysis to the spike region allowed us to minimize alignment errors and ensure more robust phylogenetic inference.

We have added this information Lines 159-162 " We focused on the spike gene because it harbors the majority of mutations responsible for immune escape and is the primary target of neutralizing antibodies and vaccines. Analyzing this region provides critical insights into reinfection risk driven by antigenic evolution.”

  1. Line 358-365, authors should also mention with reference that in global pandemic, women are generally experienced less hospitalization or mildly infection than men for which they observed very less percentage of hospitalization in their cohort.

We appreciate this suggestion. We have now added a sentence in the discussion acknowledging that globally, men have been more likely to develop severe COVID-19 and require hospitalization, which may partially explain the lower hospitalization rate observed in our cohort, especially among women.

We have included in the manuscript, Line 423-425: “Globally, women have generally experienced milder COVID-19 and lower hospitalization rates compared to men (Wu et al 2022), which may have also contributed to the relatively low proportion of hospitalizations observed in our predominantly female cohort.

  1. Authors should provide a separate paragraph explain potential reason for why chronic conditions are mostly reinfected? Is it for less immunity? But chronic conditions generally lead to more hospitalization and morbidity, but an increase in infection has not been documented. It is also possible that medication used for chronic conditions weakens the immunity. Did authors look into the viral load between chronic condition and normal infection? Assuming higher viral load in chronic condition (because they are more vulnerable and hospitalized/morbidities) may lead to higher detection rate in RT-QPCR than normal individuals which could have lower viral loads.

We thank the reviewer for this important observation. We have now added a paragraph in the discussion to explore possible explanations for the observed association between chronic conditions and reinfection. While we did not directly assess viral load due to data limitations, we discuss potential immunological and pharmacological contributors to increased susceptibility among individuals with chronic diseases.

Lines 399-408: “The increased reinfection rate among individuals with chronic conditions may be attributed to several factors. First, chronic diseases can impair immune function, reducing the ability to develop an effective response to SARS-CoV-2. Second, medications commonly used to manage chronic conditions (i.e. corticosteroids or immunosuppressants) may further weaken immune functions. In addition, it is also possible that individuals with chronic conditions have more frequent contact with healthcare services, increasing their exposure risk and likelihood of testing. While we did not assess viral load in this study, we acknowledge that higher viral loads in immunocompromised individuals could enhance RT-PCR detection sensitivity. Further studies exploring the interplay between viral kinetics, host immunity, and comorbidity burden are warranted.”

Round 2

Reviewer 1 Report

Comments and Suggestions for Authors

The authors took into account the reviewer's suggestions

Author Response

1. The authors took into account the reviewer's suggestions

We sincerely thank Reviewer 1 for their positive assessment and previous constructive feedback, which contributed to strengthening the manuscript.

Reviewer 2 Report

Comments and Suggestions for Authors

Thank you for thoroughly addressing the concerns I raised in your previous submission. 

Comments.

Avoid using the vaccine name by using solely the manufacturer's name because readers may struggle to identify which specific COVID-19 vaccine was used at the time. It may raise this problem when your manuscript is published over a decade.

For example, referring only to "Moderna" could lead to ambiguity, was it mRNA-1273 (first-generation), mRNA-1273.214 (Original/Omicron BA.1), mRNA-1273.222 (Original/Omicron BA.4/5), or mRNA-1273.815 (Omicron XBB.1.5)?

To avoid such issues, it is advisable to specify either the research code (BNT162b2, mRNA-1273, ChAdOx1-S, Ad26.COV2.S) or the commercial name (Comirnaty, Spikevax, Vaxzevria, Jcovden) rather than relying solely on the manufacturer’s name.

Author Response

1. Avoid using the vaccine name by using solely the manufacturer's name because readers may struggle to identify which specific COVID-19 vaccine was used at the time. It may raise this problem when your manuscript is published over a decade.For example, referring only to "Moderna" could lead to ambiguity, was it mRNA-1273 (first-generation), mRNA-1273.214 (Original/Omicron BA.1), mRNA-1273.222 (Original/Omicron BA.4/5), or mRNA-1273.815 (Omicron XBB.1.5)?To avoid such issues, it is advisable to specify either the research code (BNT162b2, mRNA-1273, ChAdOx1-S, Ad26.COV2.S) or the commercial name (Comirnaty, Spikevax, Vaxzevria, Jcovden) rather than relying solely on the manufacturer’s name.

We thank the reviewer for this valuable comment. To ensure clarity and long-term interpretability of the manuscript, we have revised the text to specify the vaccine formulations using either the research codes (mRNA-1273, BNT162b2, AZD1222, Ad26.COV2.S) and/or the manufacturer.

We have modified the text Lines 123-124:

(V=first vaccine, in any possible case {Ad26.COV2.S dose, BNT162b2/ mRNA-1273/ AZD1222 first dose or first + second doses

And  Table 1.

Complete vaccination schedule

A complete vaccination schedule was defined as received two doses of BNT162b2 (Pfizer-BioNTech), mRNA-1273 (Moderna) and/or AZD1222 (AstraZeneca), or a single dose of Ad26.COV2.S (Janssen) vaccine, with at least 14 days elapsed since the final dose.

Reviewer 3 Report

Comments and Suggestions for Authors

The paper is improved and may be accepted for publication

Author Response

1.The paper is improved and may be accepted for publication

We sincerely thank Reviewer 3 for their positive assessment and previous constructive feedback, which contributed to strengthening the manuscript.